# Current Take on Systemic Sclerosis Patients’ Vaccination Recommendations

**DOI:** 10.3390/vaccines9121426

**Published:** 2021-12-02

**Authors:** Giuseppe Murdaca, Giovanni Noberasco, Dario Olobardi, Claudio Lunardi, Matteo Maule, Lorenzo Delfino, Massimo Triggiani, Chiara Cardamone, Devis Benfaremo, Gianluca Moroncini, Angelo Vacca, Nicola Susca, Sebastiano Gangemi, Paola Quattrocchi, Laura Sticchi, Giancarlo Icardi, Andrea Orsi

**Affiliations:** 1Departments of Internal Medicine, University of Genova, 16132 Genova, Italy; 2Department of Health Sciences, University of Genova, 16132 Genova, Italy; noberasco.giovanni@gmail.com (G.N.); Dario.olobardi@gmail.com (D.O.); laura.sticchi@unige.it (L.S.); icardi@unige.it (G.I.); andrea.orsi@unige.it (A.O.); 3Department of Medicine, University of Verona, 37129 Verona, Italy; claudio.lunardi@univr.it (C.L.); matteo.maule@gmail.com (M.M.); lorenzodelfino1989@gmail.com (L.D.); 4Division of Allergy and Clinical Immunology, University of Salerno, 84084 Salerno, Italy; mtriggiani@unisa.it (M.T.); ch.cardamone@gmail.com (C.C.); 5Clinica Medica, Ospedali Riuniti Ancona, 60126 Ancona, Italy; devis87@gmail.com (D.B.); g.moroncini@univpm.it (G.M.); 6Dipartimento di Scienze Cliniche e Molecolari, Università Politecnica delle Marche, 60126 Ancona, Italy; 7“Guido Baccelli” Unit of Internal Medicine, Department of Biomedical Sciences and Human Oncology, School of Medicine, Aldo Moro University of Bari, 70121 Bari, Italy; angelo.vacca@uniba.it (A.V.); susnic2@gmail.com (N.S.); 8Department of Clinical and Experimental Medicine, School and Operative Unit of Allergy and Clinical Immunology, University of Messina, 98124 Messina, Italy; gangemis@unime.it (S.G.); pquattrocchi@unime.it (P.Q.); 9Hygiene Unit, “Ospedale Policlinico San Martino IRCCS”, 16132 Genova, Italy

**Keywords:** systemic sclerosis, SARS-CoV-2, influenza, *S. pneumoniae*, hepatitis, HZV, *N. meningitidis*, *H. influenzae*, HPV, diphtheria-tetanus-pertussis

## Abstract

Systemic sclerosis (SSc) is a rare autoimmune inflammatory rheumatic disease. The prevalence of SSc ranges from 7 to 700 cases per million worldwide. Due to multiple organ involvement and constant inflammatory state, this group of patients presents an increased risk of infectious diseases. This paper aimed to gather the up-to-date evidence on vaccination strategies for patients with SSc and to be a useful tool for the prevention and management of infectious diseases. The authors conducted a scoping review in which each paragraph presents data on a specific vaccine’s safety, immunogenicity, and efficacy. The work deals with the following topics: SARS-CoV-2, seasonal influenza, *S. pneumoniae*, HAV, HBV, HZV, *N. meningitidis*, *H. influenzae*, HPV, and diphtheria-tetanus-pertussis.

## 1. Introduction

Systemic sclerosis (SSc) is a rare autoimmune inflammatory rheumatic disease. The prevalence of SSc ranges from 7 to 700 cases per million worldwide [1]. Even if SSc pathogenesis is not completely clear [2] the excessive collagen production and the constant inflammatory state a can lead to multi-organ involvement and several different and serious disease presentations [3]. Apart from cutaneous involvement, SSc patients are at risk from peripheral vascular manifestations with digital ulcers and acro-osteolysis; gastrointestinal tract involvement with esophageal, gastric, and intestinal ipomobility and occlusion, malabsorption syndrome, and hepatic involvement; pulmonary involvement with interstitial lung disease (ILD) and pulmonary hypertension; heart involvement with ischemic cardiomyopathy and pericarditis; and muscular and joint involvement with myopathy and arthralgia. Apart from organ involvement, immunosuppressive therapy often represents another strong risk factor for this group of patients. Strong immunosuppressants are often used to reduce the chronic autoimmune insult, leading to a higher risk of communicable diseases. In fact, SSc mortality is the highest among rheumatic diseases [4], and infections are one of the leading causes of both hospital admission and mortality. Vaccines represent one of the safest and most effective means of disease control, but often guidelines are not on par with research evidence. This paper aimed to gather the up-to-date evidence on vaccination strategies for patients with SSc and to be a useful tool for the prevention and management of infectious diseases [5].

## 2. Materials and Methods

To identify evidence addressing field practice and available literature on vaccines, the authors decided to conduct a scoping review. Each paragraph presents data on vaccine safety, immunogenicity, and efficacy, with a specific focus on works centered on patients with SSc or similar conditions in order to provide a complete overview of the most recent recommendations.

## 3. Results

### 3.1. SARS-CoV-2 Vaccination

With more than 250 million cases, the pandemic caused by SARS-CoV-2 represents globally a great burden for communities and a great challenge for health care systems [6]. Although the infection is usually mild in the general population, it can become a great threat for people affected by comorbidities, causing increased severe clinical manifestations and death [4]. Being affected by an active disease is a significant risk factor that increases the rate of infections and the likelihood of a worse outcome [7,8]. As such patients affected by a rheumatic disease are a vulnerable group in need of special assistance [9]. Data confirmed that about half of the rheumatologic patients (46%) who contracted the infection needed hospitalization, and 10% required invasive ventilation. An increase in mortality was also determined, with a risk of 50% more in the young female population [10]. The main strategy pursued against COVID-19 has been the quick and effective development of vaccines to prevent this hard-to-treat illness. Globally, to limit the deaths caused by the pandemic, the common decision was to vaccinate the elderly and the frail population. Several studies have assessed the vaccine response reduction caused by rheumatologic patient’s therapy: corticosteroid therapy; TNF inhibitors; and anti-CD20 B-cell depleting therapy [11,12,13,14,15,16,17,18,19]. Nonetheless, the risk of contracting SARS-CoV-2 and having a far worse outcome outweighs the risk of a wasted vaccine dose or the risk of a lower response rate. This common consensus led the major regulatory organizations and scientific societies worldwide to recommend the vaccination to immunocompromised patients and patients affected by rheumatologic diseases [20,21,22,23,24,25,26,27,28]. Moreover, recent research developments have underlined the importance of a third additional vaccine dose in frail patients [29]. This differs from the booster dose now recommended to the general population.

The vaccines internationally authorized are summarized in Table 1 [30].

Due to the higher trial efficacy and the overall higher estimated effectiveness [31,32,33], mRNA vaccines should be the first recommended to frail patients.

### 3.2. Seasonal Influenza Vaccination

During the cold months of the year, up to 20% of the overall population can be infected by seasonal Influenza [34]. Influenza vaccination is strongly recommended in the majority of patients affected by autoimmune inflammatory rheumatic diseases AIIRD [35]. AIIRD are frequently complicated with infections, factors that weigh on the burden of these chronic diseases [36]. These conditions involve a higher risk of contracting influenza [37,38] and also a higher risk of influenza-related complications compared with the general population [39]. An extended report conducted on 5860 patients affected by systemic sclerosis analyzed the causes for death: among the non-SSc-related causes (41%) infections, neoplasms, and cardiovascular diseases were the most frequent (13%, 13%, 12%, respectively) [40]. In another study infections were the most common diagnoses in both overall SSc hospitalizations and SSc in-hospital mortality (17.4% and 32.7%, respectively) [41]. The pooled standardized mortality ratio is considered to be from 2.3 to 3.5 times higher than the general population [42]. Influenza epidemiological burden is estimated to be ~1 billion infections, 3–5 million cases of severe illness and 300,000–500,000 deaths yearly [35]. A literature review suggests that extra-pulmonary complications, including cardio- and cerebrovascular events and myocarditis, constitute an under-recognized burden of disease in patients infected with influenza. Influenza-specific prevention with vaccines has been shown to reduce the risk of some of these complications [43]. Inactivated influenza vaccines are recommended for high-risk individuals and showed an excellent safety profile in such patients [44], even in the ones under immunosuppressive therapy [45,46]. Furthermore, several studies showed stable disease activity and adverse event rates comparable with those of the general population after seasonal influenza vaccines [28,47,48]. The hemagglutinin inhibitor (HI) titer is a lab value used to measure influenza vaccine immunogenicity, and a value > 1:40 is considered protective [49]. The effect of the use of corticosteroids, non-biological disease-modifying antirheumatic drugs (DMARDs), and biological agents on vaccination immunogenicity has been evaluated in several studies. Influenza vaccination granted comparable levels of immunogenicity between healthy controls and patients treated with all the aforementioned therapies, except for rituximab [50,51]. Even if some studies may result in contradictory findings, the majority of patients treated with methotrexate reached adequate HI titers [52,53,54,55]. Immunogenicity is used as surrogate of on-field efficacy in almost every study. The available literature showed a lower rate of influenza attacks and bacterial complications comparing vaccinated and unvaccinated AIIRD patients [56,57]. Influenza vaccines also determined a lower rate of hospital admissions and a mortality reduction [28]. Many studies highlighted sub-optimal influenza vaccination rates among patients with AIIRD [58,59]: individuals aged >65 years old were found to have higher rates than those between 45 and 64 and those under 45 years [60]. It was also found that the vaccination coverage was higher among patients who received an active offer or a recommendation from a healthcare provider than the ones who did not. Consequently, the lack of a recommendation is found to be one of the main reasons behind non-vaccination, alongside of fear of adverse events. Given the importance of a recommendation, it is necessary to implement strategies in order to address the issue of vaccine hesitancy and increase the vaccination uptake. Strategies such as provider reminder systems, community education campaigns, home visits, financial incentives, and sensitizing general practitioners have a significant effect on vaccination rates [61]. An effective strategy that has to be considered is the introduction of influenza vaccination as a routine practice in rheumatology outpatients wards.

### 3.3. Streptococcus Pneumoniae Vaccination

Streptococcus pneumoniae represents one of the most common causes of community-acquired pneumonia (CAP) in adults, causing at least 25% of the documented cases, with bacteremia present 20% of the time [62].

Pneumococcal pneumonia (PP) and invasive pneumococcal disease (IPD) are the main causes of both morbidity and mortality among the lower respiratory tract infections (LRTIs), contributing to more deaths than all of the other studied etiologies combined [63]. The elderly population mortality risk associated with S. Pneumoniae is 3 times more than nonpneumococcal CAP [64], becoming gradually more severe in the age groups 65–69, 70–74, 75–79, 80–84, and 85–90 [65].

The National Vaccine Plan recommends the use of pneumococcal vaccines during the first year of life and after 65 years old to prevent CAP among the most susceptible age groups [66].

The main vaccines used for routine immunization are 13-valent conjugated pneumococcal vaccine and 23-valent pneumococcal polysaccharide vaccination (PPSV23).

The risk of pulmonary infection is particularly high for patients with AIIRD.

RA and SLE were identified as additional risk conditions for pneumococcal disease, with a rate of hospitalization due to pneumococcal pneumonia of 37% [67,68] and with incidence of invasive pneumococcal infection in SLE patients 13 times higher compared with the general population in another study [69].

Different studies confirmed the safety and efficacy of vaccines in individuals exposed to immunosuppressive therapies but pointed out that immunosuppressive agents reduced vaccine responses [70,71,72].

The humoral immunogenicity and safety of the PPSV23 have been demonstrated in rheumatoid arthritis (RA) [73,74,75,76,77] and systemic lupus erythematosus (SLE) [78,79,80]. A retrospective study on the long-term effect of PPSV23 in RA patients treated with MTX showed a risk of CAP almost 10 times greater in non-vaccinated patients compared with vaccinated ones. Long-term immunogenicity of PPSV23 has been evaluated in RA patients treated with methotrexate (MTX) [81] and biologics [82], showing a long-term duration of protective antibodies, up to 7 years. The immunogenicity and safety of PCV13 has been evaluated in patients with RA and was found to induce appropriate humoral response [83], although it was reduced under MTX [84]. A randomized controlled study in RA patients evaluated the serological response to PCV13 followed by PPSV23 after 16 to 24 weeks. The study demonstrated an adequate response (87% and 94% in RA patients on biologic DMARDS and conventional synthetic DMARDS), with a decreased response in patients treated with rituximab (RTX). PCV13 boosters proved ineffective at increasing the response [85]. None of the abovementioned studies raised safety concerns except for patients affected by cryopyrin-associated periodic syndromes (CAPS), who may suffer from severe local and systemic reactions after PPSV23 [86].

Patients affected by AIIRD, according to 2019 EULAR guidelines, would greatly benefit from pneumococcal vaccination. The recommended vaccination schedule according to the Centers for Disease Control and Prevention (CDC) [87,88] and the European Society of Clinical Microbiology and Infectious Diseases (ESCMID) [89] is CV13 followed by PPSV23, with an interval of at least 8 weeks between the two vaccinations.

These vaccines showed a high level of safety and efficacy in the aforementioned studies, and they should be recommended by both general physicians and specialized doctors to patients affected by AIIRD.

### 3.4. HAV Vaccination

Hepatitis A virus (HAV), a 27 nm RNA agent classified as a picornavirus, can easily infect humans, with almost 90% of children in middle- to low-income countries having an asymptomatic infection before age 10 [90]. Interhuman transmission through the fecal–oral route is the primary means of HAV transmission, particularly among close contacts, especially in households and extended family settings. Common-source outbreaks and sporadic cases can also occur from exposure to fecally contaminated food or water and from sexual behaviors considered at high risk of sexually transmitted disease, such as oro-genital sexual practices [91,92].

The infection generally has an asymptomatic course, especially in children; in some cases, a mild hepatitis may arise, with fatigue, malaise, jaundice, diarrhea, and nausea resolving within two to ten weeks. In other cases, especially in patients with comorbidities and, in particular, in subjects with chronic liver disease, the infection can also have a much more severe course with liver failure [93]. Data on incidence and prevalence of HAV infection in patients with systemic sclerosis are still lacking. Hepatitis A is a vaccine-preventable disease, and vaccination against HAV with inactivated virus should be offered to all patients considered at risk of contracting hepatitis A; such risk includes: HAV-seronegative patients travelling to or resident in endemic areas; patients having sexual behaviors at high risk of infection; patients having close contact with infected subjects and all patients with chronic liver disease, which correlates with a much more severe infection [94]. Regarding HAV vaccination efficacy, data on systemic sclerosis (SSc) are still lacking; there seems to be a strong correlation between serum antibody concentration and seroprotection. Probably, as suggested in some studies on rheumatoid arthritis (RA) and on patients on immunosuppressive drugs, a single dose of HAV vaccine could not be enough to obtain sufficient seroprotection against infection [95]. Therefore, a second booster after 6 months and determination of antibody titers is recommended [96].

### 3.5. HBV Vaccination

Hepatitis B is a potentially life-threatening liver infection caused by the hepatitis B virus (HBV). Estimates from WHO point out that almost 2 billion people have or had a serological positivity for HBV [97]. It represents an important global health problem. The epidemiology of HBV infection is very different around the world, with areas at high prevalence and areas at low prevalence [98]. In developed countries, people who frequently require blood transfusions or blood products, patients on dialysis, and recipients of solid organ transplantations are considered at high risk of contracting HBV infection, as well as intravenous drugs users, inmates, household and sexual intercourse with people affected by HBV infection, subjects with multiple sexual partners, healthcare workers, and travelers in endemic areas who have not completed their HBV vaccination [99]. Acute infection can have an asymptomatic course or, in some cases, symptoms that last several weeks, and jaundice, dark urine, extreme fatigue, nausea, vomiting, and abdominal pain may be present. A small subset of patients with acute hepatitis can develop acute liver failure, which can lead to death. In some cases, the hepatitis B virus can also cause a chronic liver infection, leading to cirrhosis and increasing the risk of liver cancer [100]. To date, the vaccine against hepatitis B, available as a single-antigen formulation or fixed combination with other vaccines (such as hepatitis A virus vaccine), is considered the best protection against chronic HBV infection and its complications, and it is included in routine childhood vaccinations in many countries. Vaccination strategies against HBV include administration of not only traditional HBsAg vaccine but also human anti-HBV surface antibody (anti-HBs), T cell vaccine, DNA vaccines, apoptotic cells expressing HBV antigens, and viral vectors expressing HBV proteins [101]. The vaccine is extremely effective, and three doses give immunity for at least 20 years. Since its introduction as a routine vaccination, the prevalence of HBV and its socioeconomic impact in industrialized countries have been greatly reduced. All people considered at high risk of infection, listed above, should be vaccinated [102]. According to the most recent studies, the prevalence of HBV in the population with autoimmune disease seems to be similar to the prevalence found in the general population [103]. Patients with SSc should receive HBV vaccination if considered at risk [104]. Data on the efficacy of HBV vaccination in these patients are still lacking; however, its safety and immunogenicity have been reported in AR patients and systemic lupus erythematosus (SLE) patients [105,106,107]. Of interest, an insufficient humoral response to HBV vaccine has been reported in patients treated with biologicals, so the determination of antibody titers may be helpful in these cases [108].

### 3.6. HZV Vaccination

Herpes zoster (HZ), also known as shingles, is a viral disease caused by reactivation of latent varicella-zoster virus in cranial-nerve or dorsal-root ganglia and is characterized by a painful, dermatomal, vesicular rash that may be complicated by post-herpetic neuralgia. In immunocompetent subjects, HZ is a common condition, especially among elderly patients, with an estimated incidence of 1 case per 100 people in the US population [109], and it is associated with substantial morbidity. Notably, immunocompromised patients can develop additional serious complications, including disseminated skin disease, verrucous skin lesions, and acute or progressive outer retinal necrosis [110]. Prevention is preferable to treatment. It is well known that herpes zoster is one of the most common viral opportunistic infections in patients suffering from autoimmune diseases, particularly patients with inflammatory myositis, SLE, RA, and inflammatory bowel disease, due also to the use of immunosuppressive drugs (glucocorticoids, DMARDs, biologic DMARDs, and targeted synthetic DMARDs) that further increase the risk of infection [98,111,112,113]. The guidelines for RA treatment recommend administering vaccination prior to initiating DMARD or biologic in patients who are 50 years old or more. Very little is known about incidence and prevalence of herpes zoster in patients with SSc, but some cohort studies have demonstrated that these patients also have a higher incidence rate of infection than general population [110]. Administration of a live-attenuated HZ vaccine (Zostavax) reduces the risk of HZ among immunocompetent individuals 50 years old and older, but it is not indicated in immunocompromised patients, who could develop a primary varicella infection [111]. Current guidelines suggest administration of a live-attenuated vaccine in patients with autoimmune disease who are considered at high risk, at least 4 weeks before starting an immunosuppressive drug. Moreover, in order to avoid a primary varicella infection, evaluation of serostatus may be considered before the administration [86]. A new non-live recombinant subunit adjuvant zoster vaccine, called Shingrix, is now available in some European countries, and it may be a candidate to replace the live-attenuated vaccine, avoiding the risk of iatrogenic infection. Furthermore, comparative studies have demonstrated a better safety profile and higher efficacy of Shingrix than the live-attenuated vaccine [112]. Shingrix is the preferred shingles vaccine in immunocompetent adults aged 50 or more, to prevent HZ and related complications with an immunization schedule of two doses, 2–6 months apart regardless of past episodes of HZ or receipt of Zostavax. Data show that the recombinant vaccine is effective in immunocompromised patients as well [113]; however, studies of safety and efficacy in patients with SSc, and in general in patients with autoimmune disease, are not yet available, and further investigation is needed. The actual recommendations on Shingrix use (ACR meeting 2020, Atlanta, GA, USA) state that all patients aged 50 with or without immunosuppressive illnesses, regardless of previous Zostavax exposure, and all patients on or starting kinase inhibitors, regardless of age, should be considered for vaccination.

### 3.7. Meningococcal Vaccination

Meningococcal vaccines are inactivated and conjugated vaccines administered intramuscularly for the prevention of infections caused by Neisseria Meningiditis. Globally each year, around one half to one million cases of invasive meningococcal disease are estimated [114].

In Italy there are three types of anti-meningococcal vaccines:The conjugate vaccine against serotype C (MenC);The tetravalent conjugate vaccine against serotypes A, C, W135 and Y (Mcv4);The conjugate vaccine against serotype B (MenB).

The Italian National Vaccine Plan proposes the administration of three doses of the anti-meningococcal B vaccine in the first year of life, whereas anti-meningococcal C vaccine or a dose of tetravalent vaccine is recommended between 13 and 15 months. One dose of tetravalent anti-meningococcal vaccine is then administered in adolescence both to those who did not get vaccinated with MenC or Mcv4 in childhood and to those who have already received a dose, since the immunoprotection is linked to a high antibody titer, which tends to decrease over time [67].

The tetravalent vaccine is recommended for all travelers going to countries where vaccine serotypes are present, such as the countries of sub-Saharan Africa. According to the Italian National Vaccine Plan, the meningococcal vaccination is strongly suggested in subjects who are considered at risk due to the presence of particular conditions: hemoglobinopathies, functional or anatomic asplenia, candidates for splenectomy, congenital or acquired immunosuppression conditions, diabetes mellitus, renal/adrenal insufficiency, HIV infection, severe chronic liver disease, cerebrovascular fluid leakage from trauma or surgery, congenital complement, Toll-like receptor 4 and properdin defects, and finally also household members of subjects with the conditions listed above. The recommendations for the administration of vaccines to scleroderma patients are the same as those identified for the entire population with systemic autoimmune diseases. The latest recommendations on vaccination in patients with autoimmune inflammatory rheumatic disease made by European League Against Rheumatism (EULAR) in 2019 do not suggest clear indications for anti-meningococcal vaccination since literature in this field is rather limited [115]. A single study in 2007 evaluated the efficacy and safety of the meningococcal C vaccine in a cohort of juvenile idiopathic arthritis patients followed for one year. In these patients, vaccination was not associated with a worsening of the underlying disease and resulted in an adequate antibody response, even in patients on immunosuppressive therapy. In general, the EULAR 2019 guidelines report that non-live vaccines, such as anti-meningococcal, can be safely administered to patients with autoimmune rheumatic disease, even in those treated with systemic immunosuppressive therapy such as glucocorticoids and DMARDs [116]. The vaccination is associated with the achievement of adequate immunoprotection, defined as the capacity to induce vaccine-specific humoral and/or cellular immune responses [117]. It is also suitable to administer vaccines during a phase of stability of the underlying disease and, if possible, before starting a planned immunosuppressive therapy; in particular, the vaccination should not be performed during B-cell depleting therapy (e.g., rituximab) since these drugs are able to inhibit the immunizing capacity of vaccines and would make vaccination ineffective. Immunocompetent household members of patients with systemic autoimmune disease should receive vaccinations, according to national guidelines [5].

### 3.8. Haemophilus Influenzae Type B Vaccination

Hib vaccination is an inactivated and conjugated vaccine administered intramuscularly for the prevention of Haemophilus influenzae type B infections. Since its introduction, the annual incidence was reduced by 99% in the US [118]. In Italy, the National Vaccine Plan schedules the immunization against type B Haemophilus influenzae during the first year (3rd, 5th, 11th month) of every infant’s life through the administration of a “combined” hexavalent vaccine, that contains six vaccines (diphtheria–tetanus–pertussis–polio–hepatitis B, and Haemophilus influenzae) in one syringe. Moreover, according to the National Plan, Hib vaccination is strongly recommended, if not previously done, in the following pathological conditions: functional or anatomic asplenia, candidates for splenectomy, congenital or acquired immunosuppression conditions, complement deficiency, subjects receiving bone marrow transplantation or awaiting solid organ transplantation, subjects undergoing chemotherapy or radiotherapy for the treatment of malignant neoplasms, and cochlear implant carriers [67]. The recommendations for the administration of vaccines to scleroderma patients are the same as those identified for patients with systemic autoimmune diseases. In 2019, the European League Against Rheumatism (EULAR) published new guidelines for vaccination in patients with autoimmune inflammatory rheumatic disease; these guidelines, which update those of 2011, do not indicate a clear schedule for Hib vaccination since literature in this field is rather limited. A recent study was carried out in Slovenia to evaluate vaccination coverage in a cohort of children with rheumatic diseases, including systemic sclerosis; at 18 years ninety-six percent of the patients experienced Hib vaccination. EULAR guidelines suggest that the administration of non-live vaccines such as Hib is reported to be safe in patients with autoimmune disease, even in patients on immunosuppressive therapy such as glucocorticoids and DMARDs. Moreover, the vaccination is judged effective and capable of inducing vaccine-specific humoral and/or cellular immune responses. Vaccinations should preferably be administered during quiescent disease and prior to planned immunosuppression, in particular B-cell depleting therapy, which is associated with a suboptimal response to vaccine. According to national guidelines, it is important to also promote vaccination in immunocompetent household members of patients with autoimmune diseases [119].

### 3.9. HPV Vaccination

Human papillomavirus (HPV) is the most common sexually transmitted infection, with a prevalence of 10% to 30% in women [120], as well as being the major cause of cervical uterine cancer worldwide [121]. Notable risk factors for cervical HPV infection are smoking, younger age at first contact, high number of sexual partners, history of sexually transmitted infections, and hormonal contraception. Whereas the majority of HPV infections are transient and spontaneously resolve within one year, high risk HPV genotypes, such as 16 and 18, may persist and be implicated in the progression of cervical dysplasia into malignancy. In general, patients with autoimmune systemic diseases are at increased risk of HPV infection, as well as pre-malignant and malignant cervical lesions [122,123,124]. Additionally, oncogenic HPV genotypes may be favored by immunosuppression, as observed in patients with systemic lupus erythematosus (SLE). Of note, most of the available evidence involves patients with SLE. In fact, in a recent systematic review, the incidence of HPV infection in patients with SSc was not shown to be significantly higher than in the general population. However, infections with multiple HPV genotypes are nearly two times more frequent in the SSc group compared with controls. Thus, limited evidence suggests that patients with SSc may have a risk of HPV infection at least comparable to the general population [35,125,126,127].

The first HPV vaccine became available in 2006. Different vaccines are nowadays marketed: a recombinant HPV quadrivalent vaccine (genotypes 6, 11, 16 and 18), a recombinant HPV bivalent vaccine (genotypes 16 and 18) [128,129,130,131,132,133,134,135], and, more recently, a recombinant 9-valent vaccine (genotypes 6, 11, 16, 18, 31, 33, 45, 52 and 58). In the US, HPV vaccination is recommended for routine vaccination at 11 or 12 years of age, although catch-up vaccination could be recommended for all subjects through age 26. In older adults, HPV vaccination provides less benefit, since more people at this age have already been exposed to HPV. In healthy young women, HPV vaccine is strongly immunogenic, inducing a high degree of protection against HPV 16/18 serotypes in bivalent vaccine and against HPV 6/11 in the quadrivalent vaccine, therefore preventing the associated premalignant lesions [136,137]. Overall, HPV vaccination has a favorable safety profile. Syncope, headache, and injection site reactions are the most commonly reported adverse events. Serious adverse events, such as adverse pregnancy outcomes, neurological disorders (including Guillain–Barré Syndrome and multiple sclerosis), anaphylaxis, venous thromboembolism, and stroke, are not associated with HPV vaccination. The immunogenicity and safety of HPV vaccine in connective tissue diseases (CTDs) has mostly been evaluated in patients with SLE. Overall, the immunogenicity of the HPV vaccine is similar in patients with SLE and controls, without significant safety concerns. Notably, vaccination does not appear to induce disease flares [138]. The recently issued EULAR guidelines state that patients with autoimmune diseases should receive HPV vaccination in accordance with recommendations for the general population. Immunogenicity and safety data of HPV vaccine in SSc patients have not been reported. Of note, after the marketing of quadrivalent HPV vaccine the adverse events reporting system recorded several new diagnoses of autoimmune diseases, including SSc and mixed connective tissue diseases. For this reason, concerns regarding the possible induction of new-onset autoimmune disease by HPV vaccine have been raised. These claims are mostly based on case reports and case series [139,140,141]. Conversely, properly designed longitudinal studies have failed to show an association between quadrivalent HPV vaccine and increased incidence of new-onset autoimmune disease [142]. With regard to SSc, in a large longitudinal observational study using administrative databases conducted in France, the incidence of SSc did not differ between young girls exposed to HPV vaccine against those non-exposed (OR 0.70, 95% IC 0.35–1.39), after a median follow-up of 33 months [143]. Several other studies confirmed the absence of an association between HPV vaccination and connective tissue diseases, even in subjects with pre-existing autoimmune diseases [144]. A recent systematic review with metanalysis confirmed these findings, observing that HPV vaccination is not associated with an increased risk of musculoskeletal or systemic autoimmune diseases (pooled OR 1.07, 95% IC 0.98–1.17) [145].

### 3.10. Diphtheria–Tetanus–Pertussis Vaccination

Diphtheria, tetanus, and pertussis are bacterial infectious diseases that may be prevented by vaccination. Their occurrence is rare for diphtheria, with less than 10 thousand cases worldwide from 2000 to 2017 [146], and tetanus, with less than 2 cases per million in Italy after 1990 [147], and more common for pertussis, estimated to have more than 2 million cases per year in the US [148]. Three different types of vaccine are available, all of which are inactivated. The DTaP and Tdap vaccines both protect against the three pathogens, whereas the Td vaccine only protects against diphtheria and tetanus. The DTaP vaccine is given to infants and young children in a series of five shots: at 2 months, 4 months, 6 months, 15 to 18 months, and again at 4 to 6 years of age. The Tdap vaccine contains lesser quantities of diphtheria and pertussis proteins, and thus it is used for booster vaccinations, such as the Td vaccine, in order to reduce the occurrence of side effects in adults [149,150]. Adults should get a dose of Tdap or Td every 10 years as well as in case of a wound that warrants tetanus vaccination. The DTaP/Tdap/Td vaccination is strongly immunogenic in immunocompetent subjects: almost 100% of healthy adults develop tetanus antibody protection upon booster immunization, whereas for diphtheria this percentage is slightly lower. Generally, tetanus toxoid and diphtheria vaccines have a good safety profile [151,152]. They can cause mild side effects such as pain or tenderness in the local area of the shot and occasionally a low-grade fever. Despite initial concerns, no relationship between these DTaP vaccination and higher grade neurological complications has been confirmed in longitudinal studies [153]. Although no studies directly assessed the efficacy and safety of DTaP/Tdap/Td vaccination in SSc patients, the EULAR guidelines state that patients with autoimmune diseases should receive toxoid tetanus vaccination in accordance with recommendations for the general population. Overall, patients with RA and SLE showed a good immunogenicity for tetanus toxoid vaccination, comparable with healthy subjects, even if treated with immunosuppressants such as rituximab (RTX) [154]. However, the efficacy of Td vaccination is assumed to be decreased upon RTX, based on data extrapolated from other vaccines [155]. Indeed, a more recent multicenter cohort study conducted in Switzerland confirmed that tetanus booster vaccination is safe and immunogenic (up to 98% of the subjects) in patients with a variety of rheumatic diseases (including RA, spondyloarthrithis, and vasculitides), whereas diphtheria vaccination may be less immunogenic (73%). In these patients, antibody response appears to be correlated with immunosuppressive therapy, particular grade of immunosuppression, more important with RTX and methotrexate, rather than the underlying autoimmune disease. Importantly, no new safety issues were observed compared with the general population [156].

### 3.11. Vaccinations in Systemic Sclerosis Patients Who Travel

Systemic sclerosis patients, especially those who take immune suppressant therapy, are more prone than healthy travelers to infectious hazards that could be encountered in a foreign country. Our group’s recommendations on this topic follow the general rules indicated for travelers that are based on the WHO recommendations for international travel and health. However, we added some additional precautions as stated in the EULAR recommendations for vaccination in adult patients with autoimmune inflammatory rheumatic diseases:

The travel should be planned in advance, in order to be scheduled when the disease activity is as low as possible, and the vaccines should be administered while taking the lowest dose of immune suppressant therapy. The aim is to keep the immunogenicity of the vaccines high and to make the patient less susceptible to infections when reaching the destination country.

Live attenuated vaccines (e.g., yellow fever) must be avoided in patients taking immune suppressant therapy. In this case, our group’s suggestion is to not travel in countries where the risk for these diseases is high. If the travel cannot be avoided, a careful discussion with the treating physician must be undertaken to consider both the potential risk of reactivation of a live attenuated vaccine and the harms of getting the disease.

We encourage patients to seek the advice of their own general practitioner or specialist to assess the disease activity and damage and to schedule with them the timing of tapering the immune suppressant therapy (if the disease course allows doing this). This should be done at least two months prior to departure. Moreover, the physician should help the patient find all the needed information about the specific infectious risk in the country of destination, the general precautions to be taken when traveling abroad, and the specific vaccination requirements demanded by the local governments. The sources of information are available on the WHO institutional site and from the patient’s own country health or foreign office. Patients should also check with their general practitioner with regard to their own vaccinal certificate to assess whether they have received the appropriate vaccinations and to assess the need for integrations. Then, the need and planning for the following vaccinations should be discussed (see WHO statements for details):−Cholera: Two live attenuated vaccines are used both with an oral route of administration. Although these vaccines have not been proven harmful for immune-suppressed patients, WHO states that cholera risk is not high for general travelers (except for emergency/relief workers); hence, careful attention to the elementary hygiene rules should be sufficient.−Dengue: A live attenuated vaccine is used. This has been proven more efficacious and less harmful in patients already seropositive for dengue, with a higher risk of severe disease when a prior seronegative vaccinated individual encounters the infection. For this reason, dengue vaccination is contraindicated in systemic sclerosis patients who take immune suppressant drugs.−Diphtheria–Tetanus–Pertussis: See section on diphtheria–tetanus–pertussis vaccine.−Haemophilus influenzae type B: See section on Haemophilus influenzae type B vaccine.−Hepatitis A: Vaccination with inactivated virus is indicated, especially in patients traveling in intermediate and high endemicity areas. For more details, see previous section.−Hepatitis B: See previous section.−Hepatitis E: Recombinant vaccine is considered for patients traveling in areas during outbreaks, especially if healthcare or relief workers. The need for the vaccination should be decided after carefully assessment of the potential risks and benefits because safety data for immune-suppressed individuals are limited.−HPV: See section on HPV vaccine.−Seasonal influenza: Inactivated vaccine is indicated before traveling in areas during the influenza season.−Japanese encephalitis: Due to the severity of infection, inactivated vaccine is indicated for patients who travel in Asia, especially during high incidence seasons and for people with long time permanence outdoor.−Measles/Mumps/Rubella/Varicella: The patient should have been vaccinated during his childhood. If not, vaccine is a live attenuated one, which should be administered with extreme caution if the patient is on immune suppressant therapy. The patient should avoid traveling in areas with active outbreak or low vaccination coverage.−Meningococcal disease: Vaccination with tetravalent conjugated or polysaccharide and recombinant group B vaccine is indicated for patients traveling in high endemicity areas, according to local serogroup prevalence. Saudi Arabia requires proof of vaccination. See specific section for more details.−Pneumococcal disease: Vaccination is indicated, especially when traveling in areas with limited access to healthcare facilities because of difficulties in the management of complicated disease. See specific section.−Poliomyelitis: Vaccination is scheduled during childhood. Oral attenuated vaccine (OPV) is contraindicated for immune-suppressed patients. Travelers from polio-free to endemic countries and vice versa should receive a full course of vaccination if not previously done, and, if already received, should have a booster dose with inactivated vaccine (IPV) 4 weeks to 12 months before departure. Some countries require proof of vaccination.−Rabies: It is fundamental to know the risk of infection for the area the patient is traveling to. For no risk areas, no vaccination is indicated. For low risk areas, vaccination with inactivated vaccine is indicated if the patient has an occupational risk (e.g., veterinarian) or a risk of contact with wild animals, especially in areas with limited access to post-exposure prophylaxis. In medium- and high-risk areas, vaccination is indicated even for patients who will spend a long amount of time outdoors, in addition to the categories mentioned before. Post-exposure prophylaxis is indicated, as per WHO recommendations.−Rotavirus: Live attenuated vaccine is contraindicated. The infection is generally mild in adults.−Tick-borne encephalitis: Vaccination with inactivated vaccine is indicated for patients who travel in endemic areas.−Tuberculosis: Calmette–Guerin bacille (BCG) is a live attenuated vaccine. Because it is a live vaccine and is not clearly efficacious in preventing tuberculosis in adults, it is not indicated for immune-suppressed patients.−Thyphoid fever: Vaccination with typhoid conjugate vaccine or injectable unconjugated Vi capsular polysaccharide is indicated for patients who travel in endemic areas.−Yellow fever: The vaccine is a live attenuated one, so vaccination is contraindicated for immune-suppressed patients. Some countries require proof of vaccination [157].

### 3.12. Vaccinations in Systemic Sclerosis Patients’ Family Members

Family members represent a contagion source for patients with systemic sclerosis, especially if they live together with them, due to close contacts and use of common household facilities. Thus, family members should be vaccinated themselves to the same pathogens to which patients with systemic sclerosis are vaccinated; also, vaccines to measles, mumps, rubella, varicella-herpes zoster, and rotavirus in 2- to 7-month-old infants should be performed. These vaccinations should be performed in conjunction with specific vaccinations indicated for the family member’s self-health. The vaccination of family members should obviously take into account their personal medical history and known contraindications to vaccination [158].

An issue to be addressed is the potential shedding of viral material with body fluids after vaccination with a live attenuated vaccine [159]. Although some studies report viral shedding with live attenuated vaccines, the vaccine recipient has not been proven to infect other people. Thus, isolation is not indicated for family members who receive a live attenuated vaccine [160]. However, there are exceptions:−Oral polio vaccine: family members should not receive this vaccine because viral shedding is infectious;−Varicella and herpes zoster: if a subject shows a varicella rash after vaccination, isolation from patients with systemic sclerosis and/or treated with immune suppressant drugs is indicated; passive immunization with varicella-zoster immune globulin could be considered;−Rotavirus: patients with systemic sclerosis should prudentially avoid contact with stool of rotavirus vaccinated patients (e.g., diapers of vaccinated infants) for four weeks, although no cases of symptomatic infection in contacts have been reported [161].

The etiology of systemic sclerosis is closely correlated with a genetic profile; thus, relatives of patients with systemic sclerosis are at increased risk of getting the disease because they also share common eventual environmental triggers [162]. In addition, cases of autoimmune diseases triggered by vaccinations are reported [163,164,165,166,167,168,169]. Thus, vaccination could be a concern for family members of patients with systemic sclerosis. Reports of localized scleroderma onset after vaccination are few and anecdotal. To our knowledge, there are no peer-reviewed reported cases of systemic sclerosis onset after vaccination. Moreover, studies have been published demonstrating no or minimal increase in disease activity after vaccination for a number of autoimmune diseases, including systemic sclerosis [170,171,172]. Thus, the possibility that vaccination could trigger a subclinical disease seems to be extremely rare, although high-quality studies to address this issue are lacking.

## 4. Discussion

Promoting prevention strategies such as vaccination is ethical, cost effective, and should be actively carried out by every physician.

Each patient should receive the appropriate means to better defend their quality of life, frail and immunocompromised patients even more so than others.

Patients affected by SSc represent the perfect example of frail, immunocompromised individuals who should receive numerous useful vaccines; but often, the literature, guidelines, and practices are not aligned. As scoped by this literature analysis, we want to point out that it should be a priority for rheumatologists and general practitioners to inform and recommend patients on the benefits and role of vaccine prevention to better control preventable communicable diseases. The principal recommendations are summarized in Table 2.

To prevent upper, lower, and severe systemic airway infections, SSc patients should receive vaccinations for the following pathogens:−SARS-CoV-2−Influenza−*S. Pneumoniae*−*N. meningitidis*−*H. influenzae*−Diphtheria-tetanus-pertussis (dTp)

Vaccination for HAV and HBV should follow the general rules of the National Plan for Vaccine Prevention. In patients older than 65 years of age, herpes zoster vaccination should be considered.

On top of those, the recent pandemic and diffusion of safe and effective vaccines calls for the use of an mRNA vaccine for every patient affected by SSc.

Table 2 shows the summary of vaccine recommendations examined in this work.

Only by having a common consensus on the management of prevention in this frail category of patients will it be possible to create a common ground for equal treatment.

## Figures and Tables

**Table 1 vaccines-09-01426-t001:** Currently authorized vaccines against SARS-CoV-2 and specific characteristics.

Vaccination	Authorization	Vaccine Technology	Cycle Schedule	Estimated Efficacy
Comirnaty—BioNTech Manufacturing GmbH	21 December 2020	mRNA	2 doses—0/21–42 days	95%
Spikevax—Moderna Biotech Spain S.L.	06 January 2021	mRNA	2 doses—0/28 days	94%
COVID-19 Vaccine Janssen—Janssen—Cilag International NV	11 March 2021	Recombinant adenovirus	1 dose	67%
Vaxzevria—Astrazeneca AB	29 January 2021	Recombinant adenovirus	2 doses—0/4–12 weeks	63%

**Table 2 vaccines-09-01426-t002:** Summary of vaccine recommendations.

Vaccination	Level of Recommendation ^1^	Considerations
SARS-CoV-2	Strongly recommended	-
Seasonal Influenza	Strongly recommended	-
*S. Pneumoniae*	Strongly recommended	-
HAV	Recommended	If clinical risk is present
HBV	Recommended	If clinical risk is present
HZV	Recommended	In the elderly
*N. meningitidis*	Strongly recommended	-
*H. influenzae*	Strongly recommended	-
HPV	Recommended	SSc does not affect vaccination choice
DTP	Strongly recommended	-

^1^ Strongly recommended, recommended, or need specific evaluation.

## Data Availability

Not applicable.

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
