# Peer review of "Current Take on Systemic Sclerosis Patients’ Vaccination Recommendations"

_vaccines, 2021, doi:10.3390/vaccines9121426_

Round 1

Reviewer 1 Report

The general impression of the article is that it is poorly written both in terms of language and content, to name few:

  • Abstract, Introduction, Materials and methods all have the same and repeated information.
  • The classification of diseases in result section is not consistent for example ,

               SARS CoV -2: - Defined in terms of epidemiology.

                Influenza: - Disease not defined.

                Hepatitis A: - Defined in terms of virus taxonomy.

                HBV: - Defined as per WHO.

               Meningococcal: - Defined based on the types of available vaccines.

  • Infectious Diseases could have been defined with the context of Ssc in Introduction rather than in Results.
  • For the language, the points I especially want to highlight are:
  1. Line 145; 10 times? 10 times decreased/low or 10 times increased/high CAP risk?
  2. Line 306 307; completely unable to understand
  3. Line 435; Terminology used “Caring physician “, I couldn’t find such term in classification of Physicians.
  4. Line 440; Preposition not used before the verb “departure”
  5. Line 554 wrong spelling is used for General Practitioner (GP)

  • In whole article the terms used for vaccines are like Inactivated, Live attenuated etc, but in line 450 authors used the word “killed bacteria” which doesn’t fall in line with articles own criteria to define the type of vaccine.
  • The discussion part is very poorly presented, I recommend to re write the whole discussion.

Apart from this the other conflicts found are: -

  • The abstract and Introduction predominantly stresses on the pulmonary infection and lung Involvement but further in the paper it is clearly not only the pulmonary infection and lung Involvement, for which the author recommends the vaccine.

  • LINE 363 and 364; author speaks about the venous thromboembolism and states there is no risk for the same with HPV vaccines, but the author had not provided any input on thromboembolism and Oxford – AstraZeneca vaccine developed for SARS-CoV-2.

  • I focused on this also for another reason because, author had described different available vaccines for all the other infectious diseases in the paper but not for SARS-CoV-2, why? my best guess here is lack of sufficient available data, but in any case, for most of the vaccines against all other infectious disease in the paper, author has clearly stated the lack of evidence.

  • LINE 84 and 85; reference 33 and 34 does not fall in the line with the given data, more reliable source of (“The pooled standardized mortality ratio is considered to be from 2.3 to 3.5 times higher than the general population”), this is required.

  • LINE 430, 437,442, 445, uses the terms he/she, it can be better written. There are patients and even audience who does not identify themselves as he or she, there are people who uses the vaccines and but does not identify themselves in category of him/ her. “Gender is not neatly divided along the binary lines of “man”/he and “woman”/she (ATLEAST IN 21st century MEDICAL PRACTISE AND SCIENCE).

Originality of the Article – None

Papers Premise – Relevant

Language and Grammar – Poor, extensive editing required.

Citation – poor

Significant contribution to the field – To some extent

Significant agreement with the current academic consensus- The paper is scientifically weak in providing the reliable and robust information (as titled), it doesn’t provide a solid proof or data which is specific for Ssc or similar disease and vaccination against infectious disease.

General Impression:

The paper can be better written, organized, and presented.

Author Response

We addressed all the main points while trying to improve English language.

  • Abstract, Introduction, Materials and methods all have the same and repeated information- expanded and defined better each section.
  • The classification of diseases in result section is not consistent for example ,

               SARS CoV -2: - Defined in terms of epidemiology.

                Influenza: - Disease not defined.

                Hepatitis A: - Defined in terms of virus taxonomy.

                HBV: - Defined as per WHO.

               Meningococcal: - Defined based on the types of available vaccines.

Infectious Diseases could have been defined with the context of Ssc in Introduction rather than in Results. – provided a classification based on epidemiology for all pathogens

  • For the language, the points I especially want to highlight are: - addressed all the points
  1. Line 145; 10 times? 10 times decreased/low or 10 times increased/high CAP risk?
  2. Line 306 307; completely unable to understand
  3. Line 435; Terminology used “Caring physician “, I couldn’t find such term in classification of Physicians.
  4. Line 440; Preposition not used before the verb “departure”
  5. Line 554 wrong spelling is used for General Practitioner (GP)

  • In whole article the terms used for vaccines are like Inactivated, Live attenuated etc, but in line 450 authors used the word “killed bacteria” which doesn’t fall in line with articles own criteria to define the type of vaccine. - rephrased
  • The discussion part is very poorly presented, I recommend to re write the whole discussion- addressed the main issues to better clarify the aims of the paper.

Apart from this the other conflicts found are: -

  • The abstract and Introduction predominantly stresses on the pulmonary infection and lung Involvement but further in the paper it is clearly not only the pulmonary infection and lung Involvement, for which the author recommends the vaccine. – reworked on both
  • LINE 363 and 364; author speaks about the venous thromboembolism and states there is no risk for the same with HPV vaccines, but the author had not provided any input on thromboembolism and Oxford – AstraZeneca vaccine developed for SARS-CoV-2. – not addressed because thought to be out of scope from recommendations
  • I focused on this also for another reason because, author had described different available vaccines for all the other infectious diseases in the paper but not for SARS-CoV-2, why? my best guess here is lack of sufficient available data, but in any case, for most of the vaccines against all other infectious disease in the paper, author has clearly stated the lack of evidence. – addressed and summarized in a table
  • LINE 84 and 85; reference 33 and 34 does not fall in the line with the given data, more reliable source of (“The pooled standardized mortality ratio is considered to be from 2.3 to 3.5 times higher than the general population”), this is required. – removed both references and provided an appropriate one
  • LINE 430, 437,442, 445, uses the terms he/she, it can be better written. There are patients and even audience who does not identify themselves as he or she, there are people who uses the vaccines and but does not identify themselves in category of him/ her. “Gender is not neatly divided along the binary lines of “man”/he and “woman”/she (ATLEAST IN 21st century MEDICAL PRACTISE AND SCIENCE).  – addressed it to make it more gender neutral

Reviewer 2 Report

The manuscript titled "Current take on Systemic sclerosis patients' vaccination recommendations" deals with an interesting topic. The authors aim gathering the up-to-date evidence on vaccination strategies for patients with Systemic sclerosis and doing a useful tool for the prevention and management of infectious diseases. Authors conducted a scoping review in which each paragraph gathers data on a specific vaccine safety, immunogenicity, and efficacy.

The authors may consider the following minor comments:

Line 136: please, specify the meaning of the abbreviations RA and SLE.

Line 145: please, specify the meaning of the abbreviation MTX.

Line 152: please, specify the meaning of the abbreviations bDMARDS and csDMARDS.

Line 153: please, specify the meaning of the abbreviation RTX.

Line 234: please, specify the meaning of the abbreviation tsDMARDS.

Line 306: please, correct infulenzae in influenzae.

Line 413: please, specify the meaning of the abbreviation SpA.

Line 564: please, write HAV with a capital v.

Table 1: HBV vaccination should be "recommended" (with additional considerations) and "not strongly recommended" based on what the authors stated in the lines 216 and 217 (Patients with SSc should receive HBV vaccination if considered at risk).

Author Response

Thanks for the insights, we address every issue and overall tried to improve the paper.

Reviewer 3 Report

Though the idea of this review article is good yet there are many points that need to be clear and solve before moving ahead.

  1. The introduction seem very small and is not accordance to the manuscript title.
  2. Throughout the text, references are absent at appropriate places. Please confirm that every sentence claimed in the manuscript should be cited properly.
  3. The data in the manuscript is not properly supported with Figures and Table data. This is hard to follow and is difficult for readers. 
  4.  Discussion section should be broaden. 

Author Response

Addressed the main points while trying to increase the overall readability.

  1. The introduction seem very small and is not accordance to the manuscript title. – addressed and expanded the introduction
  2. Throughout the text, references are absent at appropriate places. Please confirm that every sentence claimed in the manuscript should be cited properly. – provided new references and addressed the positioning issues pointed out
  3. The data in the manuscript is not properly supported with Figures and Table data. This is hard to follow and is difficult for readers.  – added tables to summarize data
  4.  Discussion section should be broadened. – tried to better address our aims

Round 2

Reviewer 1 Report

I declare that I revised this and found no conflict of interest. 

Reviewer 3 Report

Accept